# Exploring Differences in Culturable Fungal Diversity Using Standard Freezing Incubation—A Case Study in the Limestones of Lemos Pantheon (Portugal)

**DOI:** 10.3390/jof9040501

**Published:** 2023-04-21

**Authors:** Diana S. Paiva, Luís Fernandes, Emília Pereira, João Trovão, Nuno Mesquita, Igor Tiago, António Portugal

**Affiliations:** 1Centre for Functional Ecology (CFE)—Science for People & the Planet, Department of Life Sciences, University of Coimbra, Calçada Martim de Freitas, 3000-456 Coimbra, Portugal; 2FitoLab—Laboratory for Phytopathology, Instituto Pedro Nunes, Rua Pedro Nunes, 3030-199 Coimbra, Portugal; 3TERRA—Associate Laboratory for Sustainable Land Use and Ecosystem Services, Department of Life Sciences, University of Coimbra, Calçada Martim de Freitas, 3000-456 Coimbra, Portugal

**Keywords:** biodeterioration, cold, culture-dependent methodologies, freezing, fungi, limestone

## Abstract

In this study, we explored the biodiversity and abundance of culturable fungi in four samples associated with different biodeterioration outlines collected from the Lemos Pantheon, a limestone-built artwork in Portugal. We compared the results from prolonged standard freezing with those previously obtained from fresh samples to analyze differences in the obtained community and assess the effectiveness of the standard freezing incubation protocol in uncovering a different segment of culturable fungal diversity. Our results showed a slight decrease in culturable diversity, but over 70% of the obtained isolates were not present in the previously studied fresh samples. We also identified a high number of potential new species with this procedure. Moreover, the use of a wide variety of selective culture media positively influenced the diversity of the cultivable fungi obtained in this study. These findings highlight the importance of developing new protocols under varying conditions to accurately characterize the culturable fraction in a given sample. The identification and study of these communities and their possible contribution to the biodeterioration process is crucial knowledge for formulating effective conservation and restoration plans to prevent further damage to valuable cultural heritage assets.

## 1. Introduction

Environments with extreme conditions are teeming with specialized microorganisms that can survive and thrive, despite their restrictive physical and chemical conditions [1,2]. In general, rock is a very harsh substrate to colonize due to limited access to nutrients and water, temperature fluctuations, and exposure to solar radiation, among other factors [3,4]. As a result, rock dwellers are forced to withstand long periods of stress-induced inactivity, and active life is only possible during short windows of more permissive conditions [5,6,7], which makes theese microbial communities highly dynamic structures in time and space, and their species composition can vary greatly depending on these conditions [8,9,10,11]. Fungi are considered one of the most relevant stone colonizers, being highly ubiquitous heterotrophic organisms, and are able to endure extreme environmental conditions and adopt several structural, morphological, and metabolic strategies to survive [12,13].

Stonework, such as historical buildings and monuments, is present around the world, and due to its historical, cultural, artistic, and religious importance, preserving and conserving its messages and intrinsic values is critical. The role of fungi in the decay of stone monuments remained underestimated for a long time, but it is now clear that they are powerful biodeteriogens and are often responsible for severe physical, chemical, and aesthetical modifications [3,12,14,15,16,17,18,19]. Therefore, studying fungal diversity found in these stone constructions can provide comprehensive information related to biodeterioration and it is a fundamental basis for its protection [16,17]. Nowadays, the study of biodeterioration of cultural heritage is a popular subject among researchers who aim to increase their understanding of its colonizers, the interaction between these organisms and substrates, and the sensitivity to biocides, in order to set, in collaboration with restorers, protocols to prevent and control the settlement of these organisms in restoration and control programs [20,21].

Metagenomics is currently thought of as the mainstream tool in profiling complex communities in cultural heritage biodeterioration studies [22,23,24,25,26,27,28,29,30,31,32,33,34,35], but in many circumstances proved insufficient on its own. In addition, isolation provided an added value for the accurate characterization of complex fungal communities [36]. Consequently, molecular profiling can be expected to provide different results from those obtained from culture isolation techniques, and the application of both approaches is desirable to discern the biodiversity in deeper detail since they complement their outputs and tackle both intrinsic limitations [34,36]. Despite the limitations inherent to culture-based techniques (which can be time-consuming and fail to detect all fungal species present in a given sample) [37,38], they are extremely important due to the advantage of having pure cultures isolated to carry out physiological and metabolic studies, and to understand ecological functions and identify specific interactions with materials and other organisms [39,40]. Furthermore, new strains can be obtained only with isolation for further characterization and description, thus enhancing our current knowledge of the true diversity of fungi. The recovery rate of fungi is estimated to be more than 70%; therefore, culture-based approaches are still extremely useful [40]. The effectiveness of isolation techniques can vary depending on the sample and/or the environment being studied. As a result, the creation of new protocols under varying conditions is crucial.

In a previous study [41], we conducted a thorough analysis of the fungal community composition on the Lemos Pantheon, a limestone monument located in the centre region of Portugal, using high-throughput sequencing in parallel with culture-based methodologies. The combined results revealed a highly diverse and dissimilar community according to the type of biodeterioration studied. However, in a direct comparison of the two approaches, we found a dramatic divergence between the sets of fungi identified. In addition, a great number of unidentified (possibly new) taxa were recovered from selective media through cultivation, which prompted us to further explore its cultivable fungal diversity.

“Exposure of cells to suboptimal growth conditions or to any environment that reduces cell viability or fitness can be considered stresses” [42], and can be classified as biotic or abiotic, including thermal (hot or cold) and non-thermal stress (e.g., water availability) [43]. Temperature is undoubtedly one of the major factors affecting the growth and survival of any microorganism [44]. Stress tolerance responses, such as alteration in protein expression and enzymatic activity, and cytoskeletal organization, among others, can generate both immediate and long-term adaptations, which are especially crucial for the survival of organisms in environments with extreme physicochemical parameters such as stone [43,45]. Since some fungi are able to cope with different stressors, including a remarkable resilience to withstand temperatures that go beyond their typical growth range, temperature was used as a selective condition to delve deeper into the cultivable diversity. Thus, this study aimed to: (1) characterize the fungal diversity associated with different biodeterioration phenomena, present in a limestone monument, after a prolonged period of standard freezing; (2) determine if the cold stress altered the previously isolated culturable community in the samples; (3) assess the effectiveness of freezing in recovering a different segment of culturable fungal diversity, thereby enhancing our understanding of the actual culturable fraction present.

## 2. Materials and Methods

### 2.1. Site Description and Sampling Strategy

The present study was conducted on indoor samples collected in July 2021 at the Church of São Salvador da Trofa, a Catholic temple located in Trofa do Vouga, in the district of Aveiro, Portugal. This historic church houses the Lemos Pantheon, a 16th century burial place built in honor of the Lemos family, one of the most influential families in the region. Designated as a National Monument in 1992, it is the most important local landmark and a shining example of Portuguese funerary art [46]. The tomb is carved from white Ançã limestone, a unique type of Portuguese limestone, with a relatively high proportion of CaCO_3_ (>96.5%), known for its easy workability and intricate carvings [20,47]. Despite its beauty, the tomb complex has suffered from visible pathologies in certain areas, causing structural and aesthetic damage to the limestone.

In the sampling site, temperature (T) and relative humidity (RH) were monitored at the beginning and end of the sampling procedure using a digital thermo-hygrometer, with median values of T 22 °C and RH 51%. A careful observation was made, and samples were collected from areas displaying clear signs of alteration and degradation. Four samples were collected, using both micro-invasive (scalpel scraping, 0.5 g/site) and non-invasive (nitrocellulose disc swabbing, Ø 5 cm, 2/site) sampling methods, and the different types of biodeterioration observed in these areas were classified based on the ICOMOS Illustrated Glossary of Stone Deterioration Patterns: L1 indicating Dark and Green Biofilm (DGB); L2 indicating Green Biofilm (GB); L3 indicating Black Discoloration (BD); and L4 indicating Salt Efflorescence (SE) [48], as described in our previous work [41]. Each sample was divided into two aliquots, one for Illumina high-throughput sequencing and the other for direct fungal isolation (suspended in 2 mL of sterile 0.9% (*w*/*v*) NaCl solution). The results from these two analyses have already been published in Paiva et al. [41]. The samples intended for direct isolation were kept in a standard freezer at a low temperature (−18 ± 2 °C) for 6 months before they were reinoculated and used to analyze the differences in their fungal diversity composition in comparison to the fresh samples.

### 2.2. Isolation, Morphological and Molecular Identification

After being stored in a standard freezer at low temperature for 6 months, the aliquots from all 4 samples were reinoculated under the same conditions as the initial inoculation. Each aliquot was vortexed and 100 μL of the suspension was plated onto the same 6 different culture media used in the previous isolation campaign: Potato Dextrose Agar (PDA); Dichloran-glycerol Agar (DG-18); Malt Extract Agar supplemented with 10% NaCl (*w*/*v*) (MEA 10%); Czapek Solution Agar (CZ); Rose Bengal Agar Base (RB) and DSMZ 372-Halobacteria medium supplemented with 10% NaCl (*w*/*v*) (HM 10%), in triplicate. All culture media were supplemented with streptomycin (0.5 gL^−1^) to prevent bacterial growth. Inoculated media plates were incubated aerobically in the dark at 25 ± 2 °C for 6 months. The development of the colonies was checked weekly. Emerging colonies with distinct morphologies (in each culture medium) were transferred to axenic cultures in duplicate (PDA and into the medium from which they were originally recovered) and incubated for 15–30 days at the same temperature. Fungal isolates were then grouped according to their morphological traits (mycelium colour, texture and form, exudates, and growth rate), and genomic DNA was extracted from PDA pure cultures (one of each morphotype) with the REDExtract-N-Amp™ Plant PCR Kit (Sigma Aldrich, St. Louis, MO, USA). Obtained DNA was used for PCR amplification, according to the information summarized in Table 1, following the previously used protocols.

Amplification reactions were performed in 25 μL final volumes and consisted of 12.5 μL of NZYTaq Green Master Mix (NZYTech™, Lisbon, Portugal), 1 μL of each primer (10 mM), 9.5 μL of ultra-pure water, and 1 μL of template DNA, using an ABI GeneAmp™ 9700 PCR System (Applied Biosystems, Waltham, MA, USA). Visual confirmation of the overall amplification was performed using agarose gel electrophoresis (1.2%) stained with GreenSafe Premium (NZYTech™, Lisbon, Portugal) and visualized in a Molecular Imager Bio-Rad Gel Doc XR™ (Bio-Rad, Hercules, CA, USA). Obtained amplicons were purified with the EXO/SAP Go PCR Purification Kit (GRISP, Porto, Portugal) and sequenced using an ABI 3730xl DNA Analyzer system (96 capillary instruments) at STABVIDA, Portugal. Obtained sequences were analyzed and processed using Chromas v.2.6.6 (Technelysium, Southport, QLD, Australia) and deposited in the GenBank database with the accession numbers OQ211114-OQ211174 (ITS), OQ330763–OQ330787 (BenA), OQ330788–OQ330801 (CaM), OQ330802–OQ330819 (Tef1), and OQ330820–OQ330828 (Act).

Similarity searches were performed, and sequences were queried against the National Center of Biotechnology Information (NCBI) nucleotide database using a BLASTn search algorithm [56]. To ensure accurate species identification, molecular results were verified through comprehensive macroscopic and microscopic analysis of taxonomic traits, resorting to Index Fungorum (www.indexfungorum.org (accessed on 3 January 2023)) and Mycobank (https://www.mycobank.org/ (accessed on 3 January 2023)) [57,58].

### 2.3. Data Analysis

To evaluate the diversity of cultivable fungal communities resulting from freezing incubation, Species Richness (S), Shannon–Wiener index (H), Species Evenness (E), and Dominance (D) were calculated using PAST software package v.4.09 [59] for each sample (L1, L2, L3 and L4). Between samples, a Venn analysis was constructed to identify unique and shared fungal species. The community structure was analyzed based on the relative abundance of individual taxa within each sample (estimated by counting the number of emerged colonies of that morphotype in proportion to the total number of colonies for that sample). For this purpose, Microsoft^®^ Excel^®^ 2016 software was used to process, analyze, and visualize the results.

To visualize the differences in fungal diversity composition from both freezing incubation and fresh plating, ITS phylogeny was used to place retrieved isolates into their respective clades. Alignment of obtained ITS region sequences was performed using MEGA (Molecular Evolutionary Genetics Analysis) software v.11.0.11 [60] and ClustalW algorithm [61] and refined with MUSCLE [62]. Phylogenetic reconstructions were made using Maximum Likelihood (ML) analyses on the same platform. Determined by Mega software v.11.0.11, the best nucleotide substitution model for ML analysis of the obtained sequences for both cases was the Kimura 2-parameter with Gamma distribution. Bootstrap analysis was based on 100 replications.

The re-isolation of the same species can occur from different samples and different culture media plates from the same sample. Therefore, to assess fungal populations in the four studied sites, the general isolation results were analyzed as species presence–absence for each sampling site, regardless of the isolation method, as well as for each isolation media, regardless of the sampling site.

## 3. Results and Discussion

### 3.1. Frozen Diversity and Its Biodeteriorative Potential

Under the conditions used, a sum of 95 isolates was retrieved following the freezing procedure. Figure 1 displays several of the fungal isolates obtained in this study.

Based on morphology and molecular analysis of subsets of ITS, Tef, BenA, CaM, and Act sequence data, the obtained isolates were identified as 24 different species (Table A1). In addition, eleven isolates showed sufficient rDNA–ITS gene sequence divergence (similarity less than 97%) from previously cultured organisms, suggesting that they may represent novel species, although further work is required to validate their taxonomic position. Furthermore, filamentous actinobacteria belonging to the genus *Streptomyces* remained highly frequent in the samples subjected to freezing, as was observed in the fresh inoculation. Although several cultures of this bacterium were isolated, they were excluded from subsequent analysis. Diversity indices are presented in Table 2.

Overall, freezing incubation led to a decrease in diversity indices of the culturable fungi (the lowest H value in fresh samples was 1.53 in sample L4). Species richness, Shannon and Evenness index values were found to be higher in sample L1, resulting in a lower dominance, in contrast with sample L4 which presented the lowest diversity and higher dominance, indicating the presence of a more prevalent taxon (or taxa).

None of the isolated fungal species were found to be common across all four biodeterioration scenarios. Most of the species were found exclusively in one sample, with six species retrieved solely from L1 (DGB), four from L2 (GB), and seven from L3 (BD). Interestingly, L4 (SE) was the only sample without any exclusive species (Figure 2). It is plausible to suggest that this could be linked to the specific features of this sampling site, as sample L4 exhibited significant salt damage and these types of environments typically support a greater abundance of specialized organisms, which may not have the mechanisms to cope with the added stress of freezing temperatures. The L4 fresh sample contained two unique species, *Aeminium ludgeri*, a recently described xerophilic species, and a closely related unidentified isolate, which were not isolated after undergoing the freezing treatment. This suggests that these species may be sensitive to low temperatures or other stressors induced by the freezing process. Trovão et al. [63] have shown that *A. ludgeri* does not exhibit heat tolerance, and these results suggest that it may also be unable to tolerate cold and cannot survive extreme temperature variations.

Figure 3 displays the overall abundance results at the species level for all four samples. In general, *Ascomycota* was the dominant phylum in all four samples. The most frequently retrieved isolates in sample L1 were *Aspergillus protuberus* (relative abundance of 31.4%), followed by *Amycosphaerella africana* (11.6%), *Aspergillus penicillioides* (7.9%), and *Cladosporium cladosporioides* (6%); in sample, L2 were *A. Protuberus* (35.3%), *Penicillium jiangxiense* (35.3%), and *A. penicillioides* (21.6%); in sample, L3 were *C. cladosporioides* and *P. brevicompactum* (42.7% for both); and sample L4 was highly dominated by *Parengyodontium album*, which solely accounted for almost 85% of the total abundance. The relative abundance of the other *Ascomycota* species was ≤5%.

Phylum *Basidiomycota* was detected in samples L1, comprising 31.4% of the total abundance for that sample, with only one species identified, namely *Stereum hirsutum*; L2 and L3, representing less than 1% of the total abundance for those samples, with the species *S. hirsutum* and *Coprinellus micaceus* identified, respectively.

The eleven isolates for which a consensus on their identification was not attained were assigned into four different taxa, and the isolate UnId Fi_1 was found to be predominant in the samples L1 and L4 with 6.3% and 10.2% of the total abundance, respectively.

The taxonomic composition and structure of fungal communities differed among the analyzed samples, which is consistent with our previous study. Furthermore, despite being exposed to freezing temperatures, the samples yielded a wide range of isolated fungi as well as a high degree of potential novelty, confirming our earlier findings which pointed to stone as a very rich and diverse habitat [41].

To the best of our knowledge, from the 24 identified species, the following 10 were isolated (and kept in culture) for the first time, when considering limestone monuments: *A. africana*, *A. penicillioides*, *A. protuberus*, *C. allicinum*, *C. sinuosum*, *C. micaceus*, *Diaporthe foeniculina*, *P. jiangxiense*, *Talaromyces acaricola* and *Tapesia fusca* [20,34,64,65]. Excluding the unidentified isolates, many of the remaining species that were recovered have been previously related to the biodeterioration of stone and other materials. *Alternaria infectoria*, *Cladosporium* spp., *Cyphellophora olivacea* and *Stachybotrys chartarum* are known pigment producers, which have pigmented aerial mycelium and/or reproductive structures that cause chromatic alteration of the rock, an effect considered unesthetic [12,17,18]. In addition to this, *S. chartarum* can also produce extracellular pigments [66]. Besides aesthetic alterations, *A. infectoria* is also known to contribute to limestone exfoliation and biopitting [67]. Among acid-producing organisms, *Acremonium charticola*, *C. olivacea*, *P. album*, *P. Brevicompactum* and *P. crustosum* have been identified as having CaCO_3_ dissolution abilities [64,65,68,69,70,71], while *A. infectoria*, *C. cladosporioides*, *P. brevicompactum*, *P. Crustosum* and *S. hirsutum* can promote mineralization or crystallization [65,70,72,73], which can contribute to stone deterioration.

Of the species identified in this study as first reports on limestone, in general, *Aspergillus*, *Cladosporium*, *Penicillium* and *Talaromyces* species are usually widely distributed, highly ubiquitous, and are considered among the most common genera found as indoor extremotolerant contaminants [74,75,76]. The *A. penicillioides* and *A. protuberus* have been previously reported with cellulolytic and proteolytic abilities [65,69], with the latter also being able to produce extracellular pigments [77]. Curiously, in this study, abundant precipitated minerals in various shapes and sizes were observed in HM 10% culture medium surrounding the fungal mycelia of *A. penicillioides* (Figure 4). Although other cultures were grown in the same media and conditions, mineral precipitation was only observed in *A. penicillioides* cultures. To discard spontaneous mineral precipitation events, we incubated in parallel non-inoculated Petri dishes with HM 10% cultured medium, and that phenomenon was not observed.

*Aspergillus penicillioides*, an obligate halophile commonly found in hypersaline environments [78], was always found associated with biofilms (samples L1 and L2). According to Zammit et al. [79], the biofilm extracellular polymeric substances (EPS) matrix has been shown to favour the retention of water and dissolved salts, which later crystallize due to evaporation and/or desiccation, providing favourable conditions for sustaining halophile organisms. This species’ mineralization ability is likely due to the reactions of secreted acids, indicating its potential acidogenic properties. Such mineralization singularities can also promote the development of various biodeterioration phenomena [80,81,82].

*Amycosphaerella africana* and *D. foeniculina* are plant pathogens [83,84], while *C. micaceus* and *T. fusca* are saprophytic species [85,86]. The occurrence of phytopathogenic or saprophytic fungi on the surface of the limestone may be due to their presence in the surrounding vegetation and soil of the studied area. Additionally, the environmental conditions, the chemical composition of the stone, together with the presence of other organisms and/or interactions between them may provide a suitable environment for the establishment and growth of these fungi.

### 3.2. Diversity vs. Isolation Method

A maximum likelihood phylogenetic tree was constructed using rDNA-ITS sequences of one representative of each species previously identified based on morphological and taxonomic characteristics, to assess the diversity of cultivable fungi obtained from freezing incubation (Figure 5A) and compare it with that obtained from direct plating using fresh samples (Figure 5B).

The phylogenetic analysis revealed a significant difference in species composition between the two procedures. In general, low temperature restricts species diversity, as the mycobiome is less diverse when compared to direct plating. Only seven of the originally isolated species were detected after freezing, namely, *A. charticola*, *A. infectoria*, *C. cladosporioides*, *C. olivacea*, *P. album*, *P. brevicompactum* and *S. chartarum*. The stress-tolerant strategists are able to prevail even after incubation at very low temperatures. The remaining 17 species were isolated and identified for the first time only after low-temperature treatment (which represents over 70% of the total diversity obtained after the freezing treatment).

*Ascomycota* was the dominant phylum in both approaches and there was about equal representation of *Basidiomycota* species between both procedures, with only two retrieved species. At the order level, the *Cladosporiales*, *Eurotiales* and *Hypocreales* were the most common; however, the number and distribution of fungal species varied considerably. *Basidiomycota* orders were different in both cases. These results are consistent with the available literature, since the majority of fungal communities associated with low-temperature environments (e.g., Alpine, Arctic and Antarctic) are dominated by filamentous ascomycetes [87,88]. *Cladosporium cladosporioides* and some representatives of the genus *Penicillium* have been previously reported from cold environments [89,90], while the remaining species isolated in this study are being reported for the first time as cold-tolerant organisms or psychrotrophic. *Cyphellophora olivaceae* (*Eurotiomycetes*, *Chaetothyriales*) is included in the rock-inhabitant black fungi group known for its remarkable ability at exploiting all kinds of extreme environments and thriving at the edges of adaptability [91]. Therefore, its ability to resist freezing incubation does not come as a surprise. Filamentous basidiomycetes that are typically known as wood-decay fungi in temperate ecosystems have been very rarely isolated in these types of environments, [92,93] but nonetheless the species *C. micaceus* and *S. hirsutum* were only detected after freezing treatment. One very interesting and important find was the isolation of *S. chartarum* from both fresh and frozen samples. *Stachybotrys chartarum* is one of the world’s 10 most hazardous fungi when considering human and animal health, and it has been linked to sick-building syndrome [94]. Until now, this black mould has only been associated with environments that have relatively high humidity and cellulose-rich materials [95]; however, our findings suggest that this fungus may be more widespread than previously thought, as we were able to isolate it from limestone [41] and reisolate it after freezing incubation. It appears that these filamentous microfungi may be able to tolerate a broader range of conditions than previously assumed.

The impact of temperature was evident not only in the initial fungal species distribution pattern but also in their relative abundance. Although some species were common to both isolation procedures, their isolation frequencies differed significantly between samples that were directly plated versus those that were inoculated after freezing (Figure 6).

For instance, *P. album* became dominant in sample L4 since most of the species detected in direct plating disappeared, thus granting the chance for those that remained to appear more frequently. This effect is also observed in sample L3, for the species *C. cladosporioides* and *P. brevicompactum*. For these two particular samples (L3 and L4), the relative abundance of the fungal species common to both techniques is much higher when compared to the species that only appeared after cold treatment. The opposite is observed for samples L1 and L2, as they were the samples with the greatest species diversity in which the abundance of shared species decreased. As mentioned earlier in this paper, these differences may be related to the peculiarities of the sampling spot itself, as both samples originated from green biofilms.

In their natural environment, rock dwellers must cope with several different stresses and a combination of them [96]. Cold often implies osmotic stress since low temperatures increase water viscosity (ice crystals formation), and therefore cause a depletion of liquid water available for an active life. It may not be an adaptation to low temperature per se that influences fungal survival to such stress but rather to desiccation or the adaptations to both factors may be similar [89,90,96,97]. Fungi originating from polar climates, for example, have developed mechanisms to survive extreme conditions, including higher concentrations of sugars, alcohols, lipids and fatty acids, or antifreeze proteins in their cells compared to their mesophilic relatives [89,90]. Generally, hyphomycetes hyphae do not survive freezing, but spores often do [89].

The species distribution within the two protocols evidenced the predominance of well-known species, but also the presence of several relatively rare and sometimes unknown species. The seven species that were common to both methods included species belonging to cosmopolitan genera such as *Alternaria*, *Cladosporium*, *Parengyodontium* and *Penicillium* which are known for occurring in an extremely wide range of natural and man-made substrates. Their ubiquitous distribution can explain the significant prevalence of these cosmopolitan fungi even when cold stress was used. Generalist fungi are known for their high adaptability and tolerance to a wide range of physicochemical parameters, enabling them to thrive in diverse environments. Unlike specialist fungi, which are adapted to a narrow range of conditions, generalists have the ability to switch on stress responses only when required, conserving energy and resources. Despite their ability to tolerate extreme environments, generalist fungi can also be found in more common habitats, making them ubiquitous in nature. While specialists may exceed the extremotolerance of generalists in one stress factor, they often perform poorly in the absence of that specific stress and in response to other stressors [96].

### 3.3. Effect of Isolation Media

In this study, six different isolation media were used. The results revealed that fungi could be recovered from all six media, but the number of fungal isolates and the diversity of species varied among the media.

The medium MEA 10% had the most diverse group of fungal species with 12 different taxa (Figure 7A). None of the species were able to be isolated on all six media, indicating that most species had a specific preference for a certain growth medium (Figure 7B). For example, *A. infectoria*, *C. micaceus* and *T. acaricola* were only isolated from CZA; *C. sinuosum* and *P. jiangxiense* were only isolated from HM10%; *T. fusca*, *Diaporthe foeniculina* and *S. chartarum* were only isolated from MEA 10%; *C. Halotolerans* was only isolated from PDA; and *A. africana* was only isolated from RB. The remaining fungal species were isolated from more than one medium, with *A. protuberus* being the most retrieved species among different media, and DG18 being the only medium that did not have any exclusive species.

From the species segregation observed across the different media, it can be concluded that the use of various culture media was crucial for isolating a higher diversity of culturable species. Furthermore, according to their specific growth preferences, it is possible to verify a high dominance of halophilic and/or halotolerant species, distinguished by their ability to grow under high salinity conditions and subsequently low water activity. These results contrast with those of fresh samples, where xerophilic species dominated, with DG18 showing the highest number of distinct taxa in all samples, while salt-rich media, such as MEA 10% and HM 10%, had the lowest. Sterflinger’s research shed light on the possible connection between the preference for high salt media of these cold surviving fungi. In her study on “Temperature and NaCl- tolerance of rock-inhabiting meristematic fungi” [98], she observed that these fungi produce glycerol as a compatible solute to stabilize the osmotic potential under NaCl stress conditions simultaneously with temperature stress. Since glycerol accumulation is also a response to cold temperatures and acts as a cryoprotectant in fungi [89,90], this could be a possible explanation for the preference observed.

Although the cultivation and isolation of fungal species can be a challenging and time-consuming process, involving the use of different conditions and multiple culture media to ensure the highest possible diversity, as demonstrated by these results alongside those obtained from fresh samples, it is still worthwhile to make the effort when attempting to truly uncover and understand the culturable diversity. The diversity that can be obtained, the discovery of new species, and the possibility of further studying the obtained cultures, all together demonstrate the value of these classic methodologies. In fact, such efforts have led to numerous discoveries in the field of mycology and continue to be a crucial part of fungal research, making it worth the time and resources required.

## 4. Conclusions

Life finds a way to thrive in even the harshest environments. In our studies, we found that the mycobiota associated with different types of biodeterioration patterns present in limestone were taxonomically distinct, indicating that different populations occur in distinct niches on the same lithology. This finding remained valid even after subjecting these populations to cold stress. Furthermore, the freezing incubation protocol proved to be a useful tool in uncovering a different segment of culturable diversity, thereby enhancing our understanding of the actual culturable fraction present in the studied samples. By adding different culture conditions, we were able to simulate, reproduce and/or mimic the selective constraints that can naturally shape these stone mycobionts. Moreover, characterizing how selective pressures, such as freezing temperatures, favour certain species over others, and how some fungi can overcome extremely low temperatures, which can be used to predict how these fungal populations may shift according to environmental changes in their natural habitat. Since undoubtedly a minority population can have a substantial effect on the ecology, these findings are crucial knowledge to formulate effective conservation strategies to prevent biodeterioration and create effective and well-planned restoration plans.

## Figures and Tables

**Figure 1 jof-09-00501-f001:**
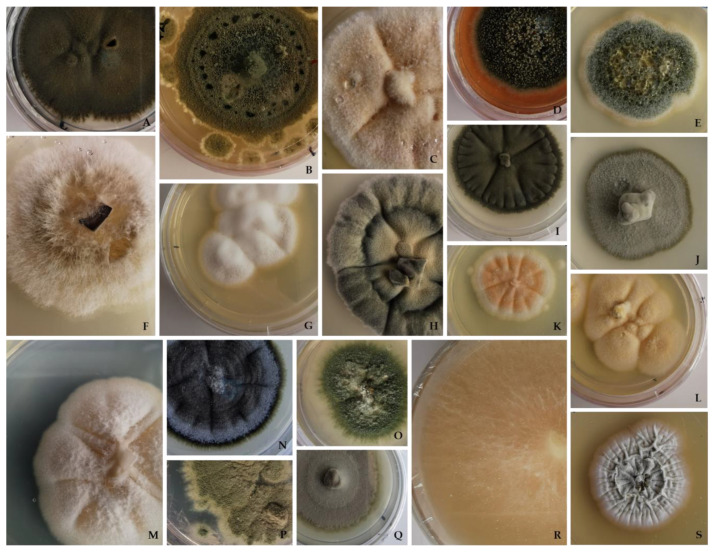
Macroscopic observations of some representatives of the fungal isolates obtained in this study on PDA medium: (**A**,**I**) *Cladosporium allicinum*. (**B**) *Penicillium crustosum*. (**C**,**E**,**L**,**O**) *Aspergillus protuberus*. (**D**) *Stachybotrys chartarum*. (**F**) *Tapesia fusca*. (**G**) *Parengyodontium album*. (**H**) *Cladosporium cladosporioides*. (**J**,**Q**) *Cyphellophora olivacea*. (**K**,**M**) *Acremonium charticola*. (**N**) *Cladosporium sinuosum*. (**P**) *Penicillium brevicompactum*. (**R**) *Stereum hirsutum.* (**S**) Filamentous *Actinobacteria* (fungus-like bacteria).

**Figure 2 jof-09-00501-f002:**
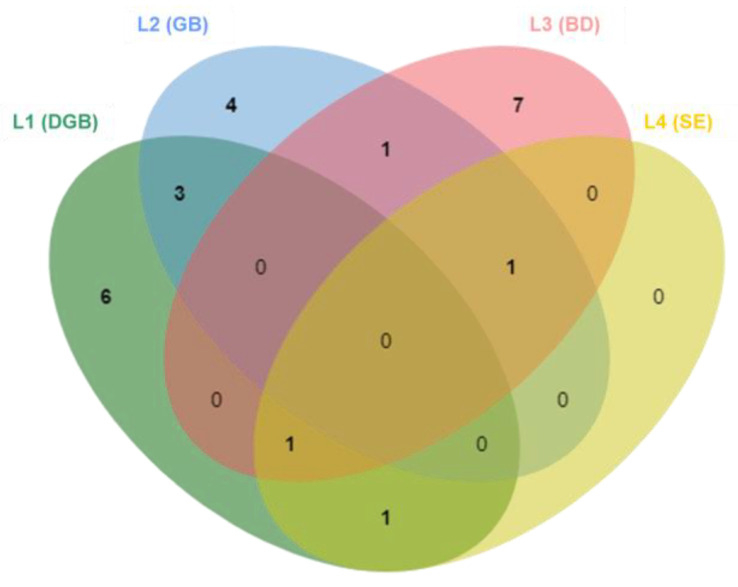
Venn diagram showing the number of shared and exclusive species between samples.

**Figure 3 jof-09-00501-f003:**
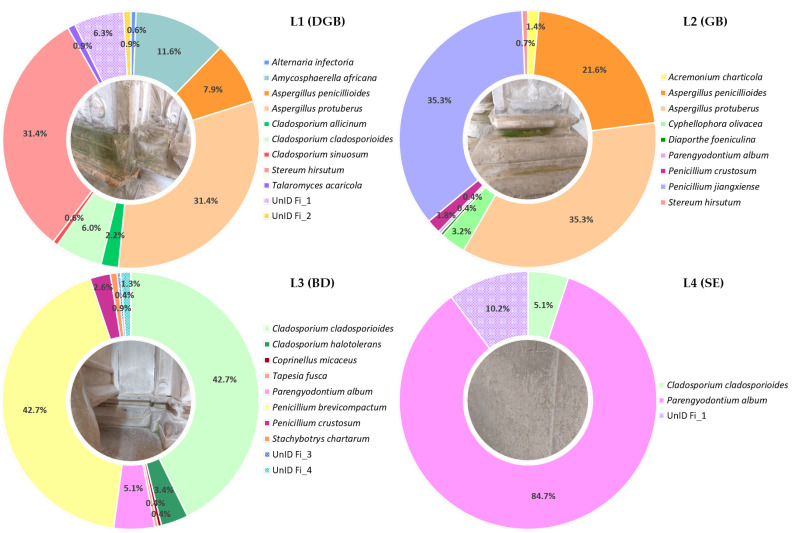
Relative abundance of the cultivable fungal diversity of all four sampling sites.

**Figure 4 jof-09-00501-f004:**
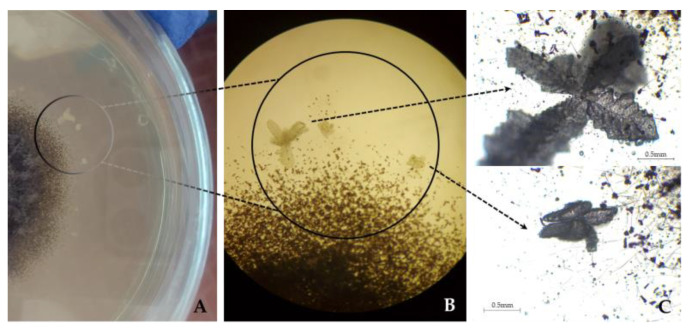
The appearance of the *Aspergillus penicillioides* colony on HM 10% medium, after maturation (gray mycelium), displaying precipitated crystals. (**A**) Direct visualization of the culture plate. (**B**) Observed under the light microscope (100×). (**C**) Observed under the light microscope (400×).

**Figure 5 jof-09-00501-f005:**
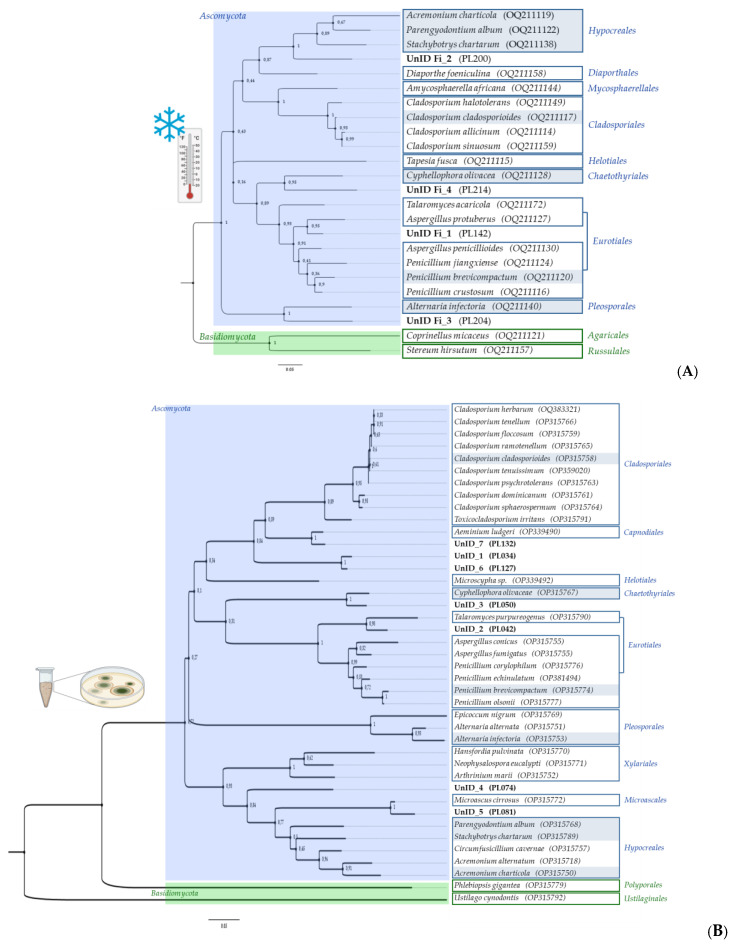
Phylogenetic trees (generated using maximum likelihood with the Kimura two-parameters model using a Gamma distribution (+G)) showing the genetic diversity among the ITS rDNA sequences of isolated fungi, obtained from: (**A**) standard freezing incubation; and (**B**) direct plating.

**Figure 6 jof-09-00501-f006:**
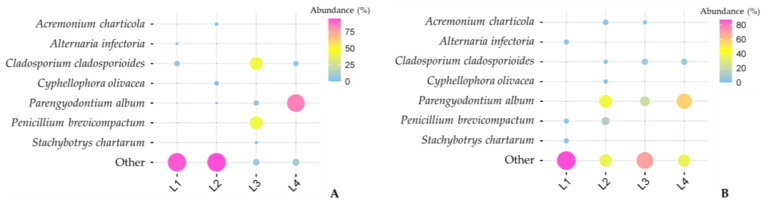
Variations in the relative abundance of shared taxa were obtained from both methods: (**A**) standard freezing incubation; and (**B**) direct plating.

**Figure 7 jof-09-00501-f007:**
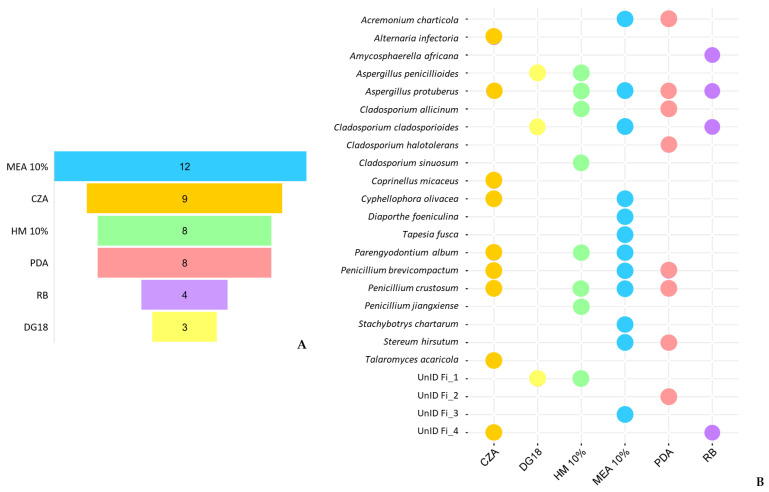
Differences in fungal isolates retrieved from the different culture media used for fungal isolation in this study. (**A**) The number of unique taxa isolated from each medium. (**B**) Overlap of cultivable species among the different culture media.

**Table 1 jof-09-00501-t001:** Primers used for fungal isolates identification.

Locus	Primers	Sequence (5′-3′)	Annealing (°C)	Amplification/Cycles	Length (bp)	Group	Reference
Internal transcribed spacer (ITS)	ITS1F ITS4	ctt ggt cat tta gag gaa gta a tcc tcc gct tat tga tat gc	52	Standard 35	~600	All Fungi	[49]
Translation elongation factor 1α (Tef1)	EF1-1018F EF1-1620R	gay ttc atc aag aac atg at gac gtt gaa dcc rac rtt gtc	56	Touchdown 36	~600	All Fungi *	[50]
β-tubulin (BenA)	Bt2a Bt2b	ggt aac caa atc ggt gct gct ttc acc ctc agt gta gtg acc ctt ggc	55	Standard 35	~500	*Aspergillus*, *Penicillium* and *Talaromyces*	[51,52]
Calmodulin (CaM)	CMD5 CMD6	ccg agt aca agg agg cct tc ccg ata gag gtc ata acg tgg	55	Standard 35	~580	*Aspergillus* and *Talaromyces*	[52,53]
Partial Actin (Act)	Act-512F Act-783R	atg tgc aag gcc ggt ttc gc tac gag tcc ttc tgg ccc at	55	Standard 35	~370	*Cladosporium*	[54,55]

* Used as a second barcode marker for all genera not specified in the table.

**Table 2 jof-09-00501-t002:** Diversity indices for cultivable fungal diversity after standard freezing incubation. The highest and lowest values are highlighted for reference.

Sample ID	Species Richness (S)	Shannon (H)	Evenness (e^H/S)	Dominance (D)
**L1**	11	1.77	0.53	0.22
**L2**	9	1.39	0.44	0.29
**L3**	10	1.27	0.35	0.37
**L4**	3	0.53	0.56	0.73

## Data Availability

All relevant data are presented in the paper. The nucleotide sequences were deposited in the GenBank database under the accession numbers: OQ211114-OQ211174 (ITS); OQ330763–OQ330787 (benA); OQ330788–OQ330801 (CaM); OQ330802–OQ330819 (tef1) and OQ330820–OQ330828 (Act).

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
