# Peer review of "Exploring Differences in Culturable Fungal Diversity Using Standard Freezing Incubation—A Case Study in the Limestones of Lemos Pantheon (Portugal)"

_jof, 2023, doi:10.3390/jof9040501_

Round 1

Reviewer 1 Report

The article “Exploring differences in culturable fungal diversity using standard

freezing incubation – A case study in the limestones of Lemos Pantheon (Portugal)” reports an investigation on the diversity of culturable fungi in 4 samples, associated with different biodeterioration outlines from a limestone-built artwork, kept in a standard freezer at -18°C for 6 months. The authors assess the fungal diversity by ribosomal ITS phylogeny and compare the results obtained for the isolated fungi from prolonged standard freezing with those obtained in a previous work from fresh samples. They find a decrease in culturable diversity but also isolates not present in the previously studied fresh samples; therefore, they consider the freezing incubation protocol a useful tool in uncovering a different segment of culturable diversity.

I suggest that authors slightly change the aims of the study, particularly point 3) “assess the effectiveness of the standard freezing incubation protocol in uncovering a different segment of culturable diversity, thereby enhancing our understanding of the actual culturable fraction present” (lines 96-98). The freezing incubation does not seem to be a very recommendable protocol to uncover a different segment of culturable diversity because it is time-consuming (at least in the conditions used in this work) and because of other aspects highlighted in the following comments. Point 3 could be changed in “determine the culturable fraction resistant to freezing” or something similar. Moreover, the choice of the freezing incubation treatment should be better explained since the samples come from an indoor environment.

It is plausible to expect that the cold stress would alter the culturable community present in fresh samples (aim 2, lines 95-96). Even if the authors invoke stress tolerance responses which involve reorganization of gene and protein expression to withstand temperatures (lines 86-91), they have not investigated such processes. Moreover, that assumes the isolated fungi were in a vegetative state when sampled from stone. On the other hand, it is plausibly that a different cold resistance of fungal spores could explain the different cultivability of some fungi in fresh and frozen samples. This can be easily tested by treating spore preparations of some fungi to few freeze/thaw cycles and then plating spores and determining the percentage of survivors. This should be done for some fungi found both in fresh and frozen samples, only in fresh samples and only in frozen samples, all fungi coming from the same sample (at least from one sample). This should be feasible if the collection of isolated fungi has been stored. Results could suggest if spore resistance is the process mainly involved.

The isolation of fungi from stone does not necessarily imply a role in biodeterioration because it can be due to their occasional presence as spores on stone. While 10 species were isolated for the first time in this work from stone, other species recovered were previously related to biodeterioration of stone and other materials. Their role, discussed by the authors, is, however, putative for the isolated strains. Since cultivation allows to study the physiology of the isolates, I suggest that authors assess biodeterioration potential of both known and unknown deteriogens for limestone by simple tests such as the acid production test.

I have some doubts about the cultivation and isolation procedure: the suspensions of the frozen samples were plated onto six different culture media in triplicate, as done in the previous work with fresh samples, and incubated aerobically in the dark at 25 ± 2 °C for 6 months. What about the physical conditions of the solid medium after few months? It is also hard to imagine the check and recovering of new emerging colonies (lines 137-138) from the plates after some weeks or few months since one or few fast-growing molds could invade most of the plate surface in few days. I kindly ask authors to better specify if they used some other incubation conditions to overcome such difficulties. Please also consider the risk of plates contamination during such a prolonged incubation.

The number of fungal isolates should be better specified: the 95 isolates retrieved following the freezing procedure (line 192) and reported in Table A1 include 10 bacteria belonging to Streptomyces, so the fungal isolates are 85. Please check if Streptomyces was included in counting the number of genera and modify it. If Streptomyces strains stay in Tab A1, this should be specified in the caption.

The authors used five phylogenetic markers to identify the isolated fungi, two (ITS and tef1) for all fungi, the other three for specific groups (Table 1). For some fungi, however, consensus identification was obtained based on culture and morphological characteristics (b in Tab A1). Moreover, for the other fungi (a, c, d in Tab A1), the accession number of tef 1 is not always reported; the authors should explain why. They should also report and comment in the text if the taxonomic assignment obtained with tef1 was in accordance with that obtained with ITS since the contribution of tef1 to the phylogeny of these fungi is not clear.

What about strain PL201? It is marked as ‘b’ although three markers were sequenced.

Why is no accession number shown for the unidentified isolates (UnID) in Tab A1?

Concerning the decrease in diversity of the culturable fungi before and after freezing, it is poorly discussed (lines 211-212). Since the aim 2 (lines 95-96) and the diversity indices were calculated in both cases, a more detailed discussion and, possibly, a quantification of the decrease in diversity are expected.

Conclusions should be rewritten considering the above comments.

Fig. 4A line 306 and Fig. 4B line 307 are Fig. 5A and Fig. 5B, respectively.The article “Exploring differences in culturable fungal diversity using standard

freezing incubation – A case study in the limestones of Lemos Pantheon (Portugal)” reports an investigation on the diversity of culturable fungi in 4 samples, associated with different biodeterioration outlines from a limestone-built artwork, kept in a standard freezer at -18°C for 6 months. The authors assess the fungal diversity by ribosomal ITS phylogeny and compare the results obtained for the isolated fungi from prolonged standard freezing with those obtained in a previous work from fresh samples. They find a decrease in culturable diversity but also isolates not present in the previously studied fresh samples; therefore, they consider the freezing incubation protocol a useful tool in uncovering a different segment of culturable diversity.

I suggest that authors slightly change the aims of the study, particularly point 3) “assess the effectiveness of the standard freezing incubation protocol in uncovering a different segment of culturable diversity, thereby enhancing our understanding of the actual culturable fraction present” (lines 96-98). The freezing incubation does not seem to be a very recommendable protocol to uncover a different segment of culturable diversity because it is time-consuming (at least in the conditions used in this work) and because of other aspects highlighted in the following comments. Point 3 could be changed in “determine the culturable fraction resistant to freezing” or something similar. Moreover, the choice of the freezing incubation treatment should be better explained since the samples come from an indoor environment.

It is plausible to expect that the cold stress would alter the culturable community present in fresh samples (aim 2, lines 95-96). Even if the authors invoke stress tolerance responses which involve reorganization of gene and protein expression to withstand temperatures (lines 86-91), they have not investigated such processes. Moreover, that assumes the isolated fungi were in a vegetative state when sampled from stone. On the other hand, it is plausibly that a different cold resistance of fungal spores could explain the different cultivability of some fungi in fresh and frozen samples. This can be easily tested by treating spore preparations of some fungi to few freeze/thaw cycles and then plating spores and determining the percentage of survivors. This should be done for some fungi found both in fresh and frozen samples, only in fresh samples and only in frozen samples, all fungi coming from the same sample (at least from one sample). This should be feasible if the collection of isolated fungi has been stored. Results could suggest if spore resistance is the process mainly involved.

The isolation of fungi from stone does not necessarily imply a role in biodeterioration because it can be due to their occasional presence as spores on stone. While 10 species were isolated for the first time in this work from stone, other species recovered were previously related to biodeterioration of stone and other materials. Their role, discussed by the authors, is, however, putative for the isolated strains. Since cultivation allows to study the physiology of the isolates, I suggest that authors assess biodeterioration potential of both known and unknown deteriogens for limestone by simple tests such as the acid production test.

I have some doubts about the cultivation and isolation procedure: the suspensions of the frozen samples were plated onto six different culture media in triplicate, as done in the previous work with fresh samples, and incubated aerobically in the dark at 25 ± 2 °C for 6 months. What about the physical conditions of the solid medium after few months? It is also hard to imagine the check and recovering of new emerging colonies (lines 137-138) from the plates after some weeks or few months since one or few fast-growing molds could invade most of the plate surface in few days. I kindly ask authors to better specify if they used some other incubation conditions to overcome such difficulties. Please also consider the risk of plates contamination during such a prolonged incubation.

The number of fungal isolates should be better specified: the 95 isolates retrieved following the freezing procedure (line 192) and reported in Table A1 include 10 bacteria belonging to Streptomyces, so the fungal isolates are 85. Please check if Streptomyces was included in counting the number of genera and modify it. If Streptomyces strains stay in Tab A1, this should be specified in the caption.

The authors used five phylogenetic markers to identify the isolated fungi, two (ITS and tef1) for all fungi, the other three for specific groups (Table 1). For some fungi, however, consensus identification was obtained based on culture and morphological characteristics (b in Tab A1). Moreover, for the other fungi (a, c, d in Tab A1), the accession number of tef 1 is not always reported; the authors should explain why. They should also report and comment in the text if the taxonomic assignment obtained with tef1 was in accordance with that obtained with ITS since the contribution of tef1 to the phylogeny of these fungi is not clear.

What about strain PL201? It is marked as ‘b’ although three markers were sequenced.

Why is no accession number shown for the unidentified isolates (UnID) in Tab A1?

Concerning the decrease in diversity of the culturable fungi before and after freezing, it is poorly discussed (lines 211-212). Since the aim 2 (lines 95-96) and the diversity indices were calculated in both cases, a more detailed discussion and, possibly, a quantification of the decrease in diversity are expected.

Conclusions should be rewritten considering the above comments.

Fig. 4A line 306 and Fig. 4B line 307 are Fig. 5A and Fig. 5B, respectively.The article “Exploring differences in culturable fungal diversity using standard

freezing incubation – A case study in the limestones of Lemos Pantheon (Portugal)” reports an investigation on the diversity of culturable fungi in 4 samples, associated with different biodeterioration outlines from a limestone-built artwork, kept in a standard freezer at -18°C for 6 months. The authors assess the fungal diversity by ribosomal ITS phylogeny and compare the results obtained for the isolated fungi from prolonged standard freezing with those obtained in a previous work from fresh samples. They find a decrease in culturable diversity but also isolates not present in the previously studied fresh samples; therefore, they consider the freezing incubation protocol a useful tool in uncovering a different segment of culturable diversity.

I suggest that authors slightly change the aims of the study, particularly point 3) “assess the effectiveness of the standard freezing incubation protocol in uncovering a different segment of culturable diversity, thereby enhancing our understanding of the actual culturable fraction present” (lines 96-98). The freezing incubation does not seem to be a very recommendable protocol to uncover a different segment of culturable diversity because it is time-consuming (at least in the conditions used in this work) and because of other aspects highlighted in the following comments. Point 3 could be changed in “determine the culturable fraction resistant to freezing” or something similar. Moreover, the choice of the freezing incubation treatment should be better explained since the samples come from an indoor environment.

It is plausible to expect that the cold stress would alter the culturable community present in fresh samples (aim 2, lines 95-96). Even if the authors invoke stress tolerance responses which involve reorganization of gene and protein expression to withstand temperatures (lines 86-91), they have not investigated such processes. Moreover, that assumes the isolated fungi were in a vegetative state when sampled from stone. On the other hand, it is plausibly that a different cold resistance of fungal spores could explain the different cultivability of some fungi in fresh and frozen samples. This can be easily tested by treating spore preparations of some fungi to few freeze/thaw cycles and then plating spores and determining the percentage of survivors. This should be done for some fungi found both in fresh and frozen samples, only in fresh samples and only in frozen samples, all fungi coming from the same sample (at least from one sample). This should be feasible if the collection of isolated fungi has been stored. Results could suggest if spore resistance is the process mainly involved.

The isolation of fungi from stone does not necessarily imply a role in biodeterioration because it can be due to their occasional presence as spores on stone. While 10 species were isolated for the first time in this work from stone, other species recovered were previously related to biodeterioration of stone and other materials. Their role, discussed by the authors, is, however, putative for the isolated strains. Since cultivation allows to study the physiology of the isolates, I suggest that authors assess biodeterioration potential of both known and unknown deteriogens for limestone by simple tests such as the acid production test.

I have some doubts about the cultivation and isolation procedure: the suspensions of the frozen samples were plated onto six different culture media in triplicate, as done in the previous work with fresh samples, and incubated aerobically in the dark at 25 ± 2 °C for 6 months. What about the physical conditions of the solid medium after few months? It is also hard to imagine the check and recovering of new emerging colonies (lines 137-138) from the plates after some weeks or few months since one or few fast-growing molds could invade most of the plate surface in few days. I kindly ask authors to better specify if they used some other incubation conditions to overcome such difficulties. Please also consider the risk of plates contamination during such a prolonged incubation.

The number of fungal isolates should be better specified: the 95 isolates retrieved following the freezing procedure (line 192) and reported in Table A1 include 10 bacteria belonging to Streptomyces, so the fungal isolates are 85. Please check if Streptomyces was included in counting the number of genera and modify it. If Streptomyces strains stay in Tab A1, this should be specified in the caption.

The authors used five phylogenetic markers to identify the isolated fungi, two (ITS and tef1) for all fungi, the other three for specific groups (Table 1). For some fungi, however, consensus identification was obtained based on culture and morphological characteristics (b in Tab A1). Moreover, for the other fungi (a, c, d in Tab A1), the accession number of tef 1 is not always reported; the authors should explain why. They should also report and comment in the text if the taxonomic assignment obtained with tef1 was in accordance with that obtained with ITS since the contribution of tef1 to the phylogeny of these fungi is not clear.

What about strain PL201? It is marked as ‘b’ although three markers were sequenced.

Why is no accession number shown for the unidentified isolates (UnID) in Tab A1?

Concerning the decrease in diversity of the culturable fungi before and after freezing, it is poorly discussed (lines 211-212). Since the aim 2 (lines 95-96) and the diversity indices were calculated in both cases, a more detailed discussion and, possibly, a quantification of the decrease in diversity are expected.

Conclusions should be rewritten considering the above comments.

Fig. 4A line 306 and Fig. 4B line 307 are Fig. 5A and Fig. 5B, respectively.

Author Response

Dear Reviewer,

We would like to thank you for your highly insightful comments and valuable criticisms of the manuscript. We have edited the original document to address your concerns. Please find attached the rebuttal letter with a point-by-point response to your comments, as well as the revised manuscript in accordance with the proposed suggestions. We hope that the revisions provided are sufficient to make our manuscript suitable for publication, and we look forward to hearing from you at your earliest convenience.

Reviewer 2 Report

The concept of this paper is highly academically value.

But some parts are missing.

I put the comment in the file.

Author Response

(The authors gave the same response as above.)

Reviewer 3 Report

The manuscript entitled "Exploring differences in culturable fungal diversity using standard freezing incubation – A case study in the limestones of Lemos Pantheon (Portugal)" (ID jof-2325441) deals with a study on diversity and abundance of culturable fungi in 4 samples associated with different biodeterioration symptoms present on Lemos Pantheon by comparing fresh and samples subjugated to freezing. The study is interesting and gives valuable insight into the physiological properties and diversity of deteriogenic fungi, well designed and the manuscript is well written. However, a few minor points (listed below) must be addressed before the manuscript is accepted for publication. 

Figure 1. The title of this figure implies that only fungal isolates are represented on it, however, Fig.1S refers to filamentous bacteria. I propose removing this photo from the Figure as the manuscript itself deals with fungal diversity only. Why are there several photos of the same fungal species? To show differences in colony morphology?

Same with Table A1. I propose removing the bacteria from the table as the name of the table implies that GenBank accession numbers of fungal isolates are given. 

Table 2. Explain what differences in color mean (gray and light gray).

H value of sample L4 is 0.53 in Table and 1.53 in the text. Correct. 

Figure 2 should be placed after it is referred to in the text. 

Figure 4. Correct B to C when referring to the last photo in the title of Figure 3.

Line 317. Replace , with a . in Cy,

When starting a sentence with a species name, write it in full and not in shortened form. 

Author Response

(The authors gave the same response as above.)

Round 2

Reviewer 1 Report

The authors have adequately addressed most of my comments. 

However, two minor points must be addressed before the manuscript is accepted for publication:

1. In their rebuttal letters, authors say that “freezing … has proven to be extremely useful in advancing the study of cultivable communities”. If previous works have demonstrated the usefulness of freezing in advancing the study of cultivable communities, that must be reported with bibliographical citation(s) since it better supports the rationale of this work.  Please add a sentence like that reported in the rebuttal after the full stop at line 93 (and before “Thus, the aims of this study were…”) with the corresponding reference(s). 

2. The caption of Table A1 is still confusing. Please change it in: “Provenance of all isolates and GenBank accession numbers of fungal isolates retrieved in this study”.

Author Response

Dear Reviewer,

We sincerely appreciate your highly insightful comments and valuable criticisms of our manuscript, which have significantly improved the overall quality of the document. We have thoroughly addressed all your concerns and revised the original manuscript accordingly. Furthermore, we would also like to clarify that our statement in the rebuttal letter regarding the usefulness of freezing in the study of cultivable communities is solely based on the results obtained in this study. Additionally, a brief introduction is provided in lines 82-90 on how fungal communities inhabiting extreme environments, such as rocks, may be affected and shaped by this temperature stress. We hope that the revisions provided are sufficient to make our manuscript suitable for publication, and we look forward to hearing from you at your earliest convenience.

Thank you again for your time and effort in reviewing our manuscript!